# Evaluation of Prognostic Factors, including Duodenal P-Glycoprotein Expression, in Canine Chronic Enteropathy

**DOI:** 10.3390/ani11082315

**Published:** 2021-08-05

**Authors:** Marco Pietra, Giorgia Galiazzo, Francesca Bresciani, Maria Morini, Sara Licarini, Maria Elena Turba, Gianmaria Amaducci, Giuliano Bettini, Federico Fracassi, Fabio Ostanello

**Affiliations:** 1Department of Veterinary Medical Sciences, University of Bologna, Ozzano Emilia, 40126 Bologna, Italy; giorgia.galiazzo2@unibo.it (G.G.); maria.morini@unibo.it (M.M.); giuliano.bettini@unibo.it (G.B.); federico.fracassi@unibo.it (F.F.); fabio.ostanello@unibo.it (F.O.); 2Veterinary Hospital H24, Firenze S.r.L., 50100 Firenze, Italy; francesca.bresciani85@gmail.com; 3Self Employed Veterinarian, 41100 Modena, Italy; sara.licarini@gmail.com; 4Genefast Srl, 47100 Forlì, Italy; me.turba@genefast.com; 5Barn Lodge Veterinary Hospital, Ormskirk L39 1LX, UK; amaduccig@gmail.com

**Keywords:** dog, chronic enteropathy, immunosuppressant-responsive enteropathy, non-responsive enteropathy, P-glycoprotein

## Abstract

**Simple Summary:**

Canine chronic enteropathies are classified according to their response to diet, antibiotics, and immunosuppressant drugs, or to a lack of response to treatment. The aim of this study was to identify which factors, including P-glycoprotein activity (a membrane-bound protein involved in multiple drug resistance), could be related to therapeutic failure. Ninety-two dogs were included in the study (73 with immunosuppressant-responsive enteropathy and 19 with non-responsive enteropathy). Factors significantly associated with an increased risk of belonging to the non-responsive group are previous treatment with glucocorticoids and hypoproteinemia. Moreover, using clonality testing, approximately one quarter of the non-responsive dogs showed a clonal response for T or B lymphocytes in the duodenal lymphocytes, although both histology and immunohistochemistry were not suggestive of lymphoma. In conclusion, additional studies are needed to investigate the underlying mechanisms that cause the non-responsive enteropathy, and whether a latent neoplastic disease could be related to the lack of response.

**Abstract:**

The aim of this retrospective single-center study was to evaluate which factors, including expression of P-glycoprotein (P-gp), a membrane-bound protein involved in multiple drug resistance, could predict the response to treatment in canine immunosuppressant-responsive enteropathy (IRE). Dogs with IRE or non-responsive enteropathy (NRE) that were examined from 2005 to 2014 were included and were divided into two groups (IRE vs. NRE). Signalment, history, and clinical and laboratory findings were collected. P-glycoprotein immunohistochemistry was carried out on duodenal biopsies of both groups stored in our biobank, and immunophenotyping and molecular clonality were performed on the NRE samples. Ninety-two dogs were enrolled, 73 IRE (79.3%) and 19 NRE (20.7%), with a prevalence of pure breed (78.3% vs. 21.7%) and male dogs *(p* < 0.001). Factors associated with a worse prognosis were previous treatment with steroids (*p* = 0.033) and lower serum total protein concentration (*p* = 0.005). Clonality testing on the NRE duodenal biopsies showed 5/16 clonal responses, assuming a latent undiagnosed lymphoma as a possible cause of the NRE.

## 1. Introduction

In dogs, chronic enteropathies (CEs) are characterized by persistent gastrointestinal signs and are phenotypically classified into three categories: (1) food-responsive enteropathy (FRE); (2) antibiotic-responsive enteropathy (ARE); and (3) immunosuppressant-responsive enteropathy (IRE). Moreover, a small percentage of dogs do not respond to any treatment: they are classified as patients with non-responsive enteropathy (NRE) [1]. Another category out of this classification is protein-losing enteropathy (PLE), a syndrome characterized by enteric protein loss with consequent hypoalbuminemia which is associated with a worse prognosis [2,3].

As highlighted by the classification, the diagnosis of CE is based on response to treatment. In fact, the severity of signs, laboratory findings, and endoscopic and histopathologic procedures are generally not enough to identify the phenotype of the enteropathy [4]. In dogs with CE, especially with the IRE phenotype, the response to treatment influences the dog’s quality of life, the time of survival, the economic impact on owners, and, sometimes, the decision for euthanasia.

Several researchers have focused on studying the possibility of finding useful markers for predicting the response to treatment and the prognosis of dogs with CE. Until now, a high clinical activity index (canine inflammatory bowel disease activity index—CIBDAI, or canine chronic enteropathy clinical activity index—CCECAI) [5,6], hypocobalaminemia [7], hypoalbuminemia [2], high serum C-reactive protein concentration [5], high canine pancreatic lipase immunoreactivity concentration [8], and an elevated neutrophil-to-lymphocyte ratio [9] in IRE dogs have been identified as conditions correlated with a negative outcome with failure of treatment response. The role of P-glycoprotein (P-gp), concerning the failure in the response to treatment in CE, has been studied in recent years [10,11]. P-glycoprotein is a membrane-bound protein encoded by the ABCB1 gene, which acts as a non-specific efflux pump capable of removing drugs, and is responsible for multiple drug resistance (MDR) [12,13]. Drugs extruded from the cell by P-gp include steroids, immunosuppressive drugs (e.g., cyclosporine), and antimicrobial agents (e.g., doxycycline, erythromycin) as well as anticancer agents (e.g., vinca alkaloids, doxorubicin), and others [14]. Its presence has been identified on the membrane of lymphocytes and on the brush border membrane of enterocytes as well as on the membranes of other cells (e.g., biliary duct cells, proximal cells of the renal tubules, cells of the adrenal glands, and choroid plexus epithelial cells), both in humans and in dogs [10,11,15].

One of the most studied substrates of P-gp is glucocorticoids; in human patients with inflammatory bowel disease (IBD), the expression of P-gp in the peripheral T-lymphocytes, intra-epithelial lymphocytes, and *lamina propria* lymphocytes appears higher in individuals with enteritis unresponsive to steroids, rather than in those with steroid-responsive enteritis [14,16].

Similarly, in veterinary medicine, the relationship between the expression of P-gp in duodenal samples of dogs with CE and the lack of response to treatment with steroids has been demonstrated by Allenspach et al. [10], who showed how lower levels of P-gp expression in duodenal *lamina propria* lymphocytes before treatment correlated with a good prognosis. However, it is unclear whether changes in duodenal P-gp expression in dogs with CE can be defined as a predisposing factor capable of reducing the effectiveness of the treatment, or if they represent a consequence of previous treatment with immunosuppressive drugs [10,11].

Based on these previous results, the aim of this retrospective single-center trial was to: (1) evaluate which factors could predict the response to treatment in dogs with IRE; (2) evaluate whether the lack of response could be related to the duodenal epithelial and *lamina propria* lymphocyte expression of P-gp; (3) evaluate whether a previous treatment with steroids could induce higher expression of duodenal P-gp; and (4) differentiate inflammatory disease vs. lymphoma in duodenal samples of NRE dogs by immunophenotypic evaluation of the lymphocyte infiltrate and clonality testing.

## 2. Material and Methods

### 2.1. Inclusion Criteria and Analyzed Parameters

The electronic medical records of the Teaching Veterinary Hospital (University of Bologna, Ozzano dell’Emilia (BO), Italy) were searched for client-owned dogs, who were examined from January 2005 to December 2014, diagnosed with IRE, following the Dandrieux [1] criteria, and treated with immunosuppressive drugs (corticosteroids, cyclosporine, azathioprine).

Dogs already treated with immunosuppressive drugs, as therapy for chronic enteropathies, and referred for a second opinion, were excluded.

Signalment, comprehensive of the breed, sex, age, and weight at inclusion, was obtained from the records.

To assess the possible influence of sex and breed on the development of the disease, all dogs which were first presented to the Veterinary Hospital between 2005 and 2014, excluding subsequent visits, were considered to be the control group.

By means of a telephone call investigation, carried out at least one year after the diagnosis of the last cases included, some information regarding the follow-up, encompassing clinical condition, treatment utilized, time of treatment suspension, relapse after immunosuppressant drug discontinuation, survival time, and causes of death or suppression, were collected.

Based on these data, the dogs were divided into two groups:Group A (responsive): included (1) dogs without gastrointestinal signs, alive at the time of the telephone call or which had died from other reasons, which had not received any therapy (except diet) for at least six months; (2) dogs without gastrointestinal signs, alive at the time of the telephone call or which had died (from other reasons), which had constantly been treated with immunosuppressive drugs, tylosin, and diet; and (3) dogs which had relapsed after treatment withdrawal, and were again treated, again without gastrointestinal signs, with immunosuppressive drugs, tylosin, and diet.Group B (not responsive): included dogs which did not respond to diet, antimicrobials, or immunosuppressive drugs, dogs, which had died spontaneously or had been euthanized for related causes, or dogs with signs of NRE which had died from other causes.

Additional information, if any, acquired from the clinical database, was as follows:Remote and recent history before the inclusion, considering the previously utilized diet (commercial, home-made, or mixed diet), previous enteric parasitosis, previous protozoal and/or canine parvovirus infection, and previous treatment with glucocorticoids in the last year, for diseases not related to enteropathy, if administered for more than 10 days;Clinical signs reported at the time of inclusion, such as decreased appetite, weight loss, vomiting, diarrhea, and presence of ascites or peripheral edema;Clinical score (CCECAI) at inclusion;Laboratory findings at inclusion, encompassing: packed cell volume (PCV %), platelet count/μL, white blood cells/μL, serum cholesterol (mg/dL), serum albumin (g/dL), serum total protein (g/dL), serum folate (μg/L), and serum cobalamin (ng/L).

### 2.2. Immunohistochemical Techniques

Replicate sections of duodenal biopsy samples of both groups, collected during the diagnostic endoscopic procedures, fixed in 10% neutral buffered formalin and embedded in paraffin, were obtained from each block and selected for P-gp immunohistochemistry (IHC).

An additional immunophenotypic evaluation of the lymphocyte infiltrate was carried out on the duodenal biopsy samples of Group B’s dogs.

Tissue sections were routinely deparaffinized in xylene and rehydrated in graded ethanol. After incubation with 3% hydrogen peroxide in methanol for 30 min (for endogenous peroxidase inhibition), the samples were incubated with citrate buffer pH 6.0 in a microwave (750 W) 2 × 5 min for antigen retrieval. The sections were incubated overnight at 4 °C in a humid chamber with the primary antibody (1:1.500 monoclonal mouse anti-P-gp clone C494, Biolegend, San Diego, CA, USA) diluted in blocking solution (10% normal goat serum in phosphate-buffered saline, PBS). The sections were then washed in PBS and incubated first with the secondary antibody (1:200 goat anti-mouse IgG biotinylated, in blocking solution) for 30 min at room temperature in a humid chamber and then with streptavidin-peroxidase complex (DAKO, Glostrup, Germany) for 25 min at room temperature. After 12 min of incubation with DAB chromogenic solution (diaminobenzidine 0.02%, and H_2_O_2_ 0.001% in PBS), the sections were rinsed with PBS and then with running tap water, counterstained with hematoxylin, dehydrated, and mounted with permanent mounting medium DPX (Fluka, Riedel-de Haen, Germany). A pathologic canine liver was used as a positive control for the immunohistochemical procedure. As a negative control, 10% normal mouse serum was used instead of the primary antibody.

A semi-quantitative immunohistochemical assessment was carried out, considering cytoplasmic immunostaining on the entire tissue (cross-section) for each case. The P-gp expression was separately evaluated in both the *lamina propria* infiltrating lymphocytes and in the epithelial cells.

To evaluate the *lamina propria* infiltrating lymphocytes, ten areas of 10,000 µm^2^ were counted, five areas in the villi and five in the crypts.

For each area, the positive lymphocytes were counted and the median value was calculated; the P-gp score used was that proposed by Allenspach et al. [10]: no positive lymphocytes seen in any area (score 1), 1–4 positive cells (score 2), 5–10 positive cells (score 3), and >10 positive cells (score 4).

The positivity of the epithelial cells of the duodenal mucosa was evaluated in the same fields with the modified semi-quantitative scoring system proposed by Van der Hayden et al. [11]: score 1 if no immunopositive P-gp epithelial cells were seen in any area, score 2 for P-gp expression by epithelial cells at the level of the villus tips, and score 3 for continuous P-gp immunolabelling at the brush border of the surface epithelium.

For the immunophenotypic evaluation of the lymphocyte infiltrate in Group B, serial sections of the formalin-fixed/paraffin-embedded samples of the same section of duodenum examined for P-gp expression were deparaffinized and rehydrated for IHC, to detect the expression of CD3 (1:10 dilution; Leukocyte Antigen Lab, P. Moore, UC Davis, Davis, CA, USA) in the T cells, and CD79a (1:400 dilution; mouse monoclonal, Santa Cruz Biotechnology Inc., Dallas, TX, USA) in the B cells. The protocol used was the same as that reported for P-gp.

Samples of normal canine lymph nodes were used as positive controls in order to assess the specificity of the reactions and ensure adequate cross-reactivity in the dogs. For negative controls, the primary antibodies were replaced with homologous nonimmune sera.

### 2.3. Molecular Analysis

Genomic DNA was extracted from the duodenal samples of Group B, using a commercial kit (Maxwell^®^ RSC DNA FFPE Kit, Promega Italia, Milano, Italy) with an automatic instrument (Maxwell RSC, Promega Italia, Milano, Italy). The DNA was then PCR amplified targeting B-cell receptor (BCR) and T-cell receptor (TCR) genes using the oligonucleotides previously described [17,18,19,20,21]; a total of five different PCRs for BCR and seven different PCRs for TCR were respectively amplified. The PCR products underwent GeneScanning analysis on an automated sequencer (ABI 310, Life Technologies Italia, Monza, Italy). The electropherograms were then interpreted by an independent analyst; all the PCR products were run in duplicate and ambiguous patterns were confirmed by repeating the PCRs. The clonality patterns were interpreted as clonal (including the clonal variation on a polyclonal background), polyclonal (including the polyclonal variation with minor clones), and oligoclonal according to Keller et al. [22]. Samples having one or two sharp peaks in duplicate were classified as clonal; samples showing a Gaussian distributed curve were classified as polyclonal, and samples with three to five reproducible peaks were classified as oligoclonal. The clonal samples were interpreted as being the most indicative of lymphoma, whereas oligoclonal and polyclonal samples were considered to be consistent with a reactive population.

### 2.4. Statistical Analysis

The Kolmogorov–Smirnov test (K-S) for goodness of fit was used to verify normality of the quantitative data. Preliminarily, the composition by sex, weight, and age of the two groups (A, responsive and B, non-responsive) was compared by using the χ^2^ test and the Mann–Whitney (M-W) U test. The crude odds ratio (OR) and the 95% confidence interval (95% CI) were calculated to evaluate associations between IRE and breed and sex. The general control population used was that of mixed breed dogs admitted to the Teaching Veterinary Hospital (University of Bologna, Italy) from 2005–2014. For the breeds more represented (n > 3), the control population consisted of both male and female dogs of the same breed, admitted during the same period.

Responsive or non-responsive status was assessed as a predictor variable for monthly dog survival over the 10-year duration of the study using Kaplan–Meier survival estimates and was then compared for equality with a log-rank test.

The P-gp scores (in both the *lamina propria* infiltrating lymphocytes and the epithelial cells) and the CCECAI score were dichotomized based on the median values (3, 1, and 5, respectively). Subsequently, a two stage-analysis was carried out. In the first stage, the categorical variables were screened using the χ^2^ test. In the second stage, the factors which were screened through (*p* < 0.10) were evaluated using unconditional multiple logistic regression, carried out with SPSS 26.0.0 (SPSS Inc., Chicago, IL, USA). The goodness of fit of the models was assessed based on the likelihood-ratio statistic and the Hosmer–Lemeshow statistic. Two models were built: one based on the simultaneous entry of all the variables and the other on the reduction procedure (backward stepwise, with removal based on the likelihood-ratio statistic and *p* < 0.05). Before the multiple logistic regression, the correlation matrix was examined for potential collinearity among the independent variables. No variable was highly correlated (highest correlation coefficients = 0.59 and 0.39, all the other coefficients were lower than 0.25).

## 3. Results

A total of ninety-two dogs were enrolled in the study. Twenty dogs (21.7%) were mixed breed, while 72 (78.3%), were pure breed: German Shepherd (n = 13), Boxer (n = 12), Labrador (n = 5), Maltese, French Bulldog, Jack Russell Terrier, Beagle, Border Collie, Miniature Pinscher, Rottweiler, Shih-Tzu, Siberian Husky (two animals for each breed), American Bulldog, American Staffordshire Terrier, Great Dane, Alaskan Malamute, Bolognese, Yorkshire Terrier, Dalmatian, Basset Hound, Miniature Dachshund, English Bulldog, Cocker Spaniel, Cane Corso, Argentino Dogo, Dogue de Bordeaux, Drahtaar, Golden Retriever, Pekinese, Pointer, English Setter, Irish Setter, Springer Spaniel, Volpino Italiano, West Highland White Terrier, and Maremmano Abruzzese Sheepdog (one animal for each breed). There were 61 intact males and 2 neutered males (68.5%), 16 intact females and 13 spayed females (31.5%).

Group A (responsive) consisted of 73 dogs (79.3%) and Group B (not responsive) consisted of 19 dogs (20.7%). Overall, the median ages (min-max) at diagnosis of the dogs in Group A and B were 37 (7–168) and 46 (9–165) months, respectively. The median body weights (min-max) for the dogs in Group A and B were 24.2 (3–63), and 19.5 (5.6–43.9) kg, respectively. No significant differences were observed among the groups when comparing sex (χ^2^ = 1.24; *p* = 0.280), age (M-W U test: 559.00; *p* = 0.194), and body weight (M-W U test = 587.00; *p* = 0.304).

Overall, the odds of IRE/NRE development were 2.17 times higher among male patients than female (95% CI: 1.40–3.37; *p* < 0.001) (Table 1).

In German Shepherds, Boxers, and Labradors, the odds of IRE/NRE development were significantly higher (*p* < 0.001) than in the mixed breed control group (respectively, OR = 5.75, 95% CI: 2.86–11.58; OR = 10.67, 95% CI: 5.21–21.87; OR = 13.77, 95% CI: 5.14–36.83) (Table 1).

Within German Shepherd dogs, Boxer, Labrador, and mixed breed, no significant differences (*p* > 0.05) in the odds of IRE/NRE development were observed between gender (OR = 2.34, 95% CI: 0.64–8.53; OR = 1.64, 95% CI: 0.49–5.48; OR = 8.47, 95% CI: 0.47–154.11; OR = 2.02, 95% CI: 0.80–5.06, respectively) (Table 1).

Survival time for the dogs in Group A (1173 ± 808 days) was significantly longer than for the dogs in Group B (528 ± 596 days) (Mantel–Cox log-rank test = 12.39; *p* < 0.001) (Figure 1).

Univariate analysis of the historical data showed a significantly (*p* < 0.05) lower proportion of dogs in Group A, with respect to dogs in Group B, as regards the following parameters: previous treatment with glucocorticoids and decreased appetite (17.8% vs. 52.6% and 24.7% vs. 52.6%, respectively), while vomiting was more represented in dogs of Group A (54.8% vs. 26.3 *p* < 0.05) than in dogs of Group B.

Regarding packed cell volume (PCV), the proportion of dogs with normal, below, or above value was statistically different (*p* < 0.01) between the groups. In particular, in Group A, 81.8% of dogs had normal values (66.7% in Group B), 3% decreased (27.8% in Group B), and 9.0% increased (5.6% in Group B) (Table 2).

The proportion of animals with normal serum cholesterol, serum albumin, and serum total protein concentration values was significantly higher (*p* < 0.01) in Group A than in Group B. No significant differences (*p* > 0.05) were observed between the two groups for the other historical data examined (diet, previous enteric parasitosis, protozoal and/or canine parvovirus infection), clinical signs at inclusion (weight loss, diarrhea, ascites or peripheral edema), clinical score (CCECAI), and laboratory findings (platelet count, white blood cells, serum cobalamin, serum folate, P-gp score in *lamina propria* infiltrating lymphocytes, and P-gp score in the epithelial cells) (Table 2).

The P-gp immunohistochemical evaluations made it possible to assign a score in all the cases under examination. Representative images of these results are illustrated in Figure 2. In the lymphocytes of the *lamina propria*, the P-gp immunopositivity, when present, was always intense and cytoplasmic; in the epithelial cells, when manifested, it appeared to be focal to diffuse and with mild to strong intensity, with occasional reinforcement on the apical luminal portion of the duodenal epithelial cells (Figure 2a,b,f).

Statistically significant factors (*p* < 0.10) identified in the univariate analysis (previous treatment with glucocorticoids, decreased appetite, weight loss, vomiting, ascites or peripheral edema, PCV, serum cholesterol concentration, serum albumin concentration, serum total protein, and P-gp score in the *lamina propria* infiltrating lymphocytes) were evaluated using multiple logistic regression.

The results of the multiple logistic regression are reported in Table 3. The data were obtained from the backward stepwise model, with removal based on the likelihood-ratio statistic and *p* < 0.05.

Factors significantly associated with an increased risk of a dog belonging to the non-responsive group (Group B) were a previous treatment with glucocorticoids (OR = 8.06; *p* = 0.033) and a serum total protein value < 5.6 g/dL (OR = 37.33; *p* = 0.005). Values of a PCV > 55% and a P-gp score in the *lamina propria* infiltrating lymphocytes ≥ 3 were not significantly associated (OR = 31.20; *p* = 0.055 and OR = 1.97; *p* = 0.077, respectively). The Hosmer–Lemeshow statistic and the likelihood-ratio statistic had *p* values of 0.67 and <0.001, respectively.

No association between the P-gp score in the duodenal epithelial lymphocytes/*lamina propria* lymphocytes and eventual treatment with glucocorticoids in the previous year was found (*p* = 0.37).

Appendix A report the historical, clinical, and laboratory findings, CCECAI scores, the P-gp score in the *lamina propria* infiltrating lymphocytes, and P-gp score in the epithelial cells, of Group A and B dogs, respectively.

The results of immunophenotypic evaluation of the lymphocyte infiltrate, carried out on the biopsy samples of the dogs (16/19) in Group B, showed a mixed T-cell (CD3 immunopositive) and B-cell (CD79 immunopositive) population in the *lamina propria*, and an intraepithelial T-cell infiltration (Table 4). Neither the T- nor the B-cells exhibited atypia or an increased mitotic rate, and IHC therefore suggested the inflammatory nature of the enteritis.

On clonality testing, carried out on the same biopsy samples used for the IHC, the overall extracted DNAs were of good quality and quantity and PCR products were present for all the samples analyzed. Combined clonality results for BCR and TCR of the same sample were indicated in Table 4. Overall, a total of 5 different BCR and TCR gene rearrangement combinations were seen: 9 cases (56.25%) were polyclonal for BCR and TCR; 2 cases (12.5%) resulted in being polyclonal for BCR and oligoclonal for TCR; 4 cases (25%) were polyclonal for BCR and clonal (or its variant clonal on a polyclonal background) for TCR; 1 case (6.25%) was clonal, and its variant clonal on a polyclonal background, for both BCR and TCR, respectively.

## 4. Discussion

Immunosuppressant-responsive enteropathy treatment represents a long and complex procedure with important economic repercussions for the owner. Therefore, risk assessment and the prediction of response to treatment at the time of diagnosis would represent a useful element for the veterinarian, who could opt, among the possible immunosuppressant drugs which can be utilized (prednisolone, cyclosporine, azathioprine, chlorambucile, etc.), for drugs with greater or weaker activity, even if they have different costs and side effects. 

The work carried out in this study has made it possible to examine, over ten years, the follow-up of patients with immunosuppressant-responsive enteropathy or non-responsive enteropathy in order to find useful elements for predictive purposes.

First, in the present research, the percentage of dogs with enteropathy which were non-responsive to treatment as compared to those who responded, did not differ from what has already been reported in the literature. In fact, in the present case, 20.7% of the patients with IRE did not respond to treatment, with a range similar to that already documented by some authors (18% [6], and 5–27% [23]).

Regarding breed predisposition for developing the disease, the analysis of the present data identified some breeds (German Shepherds, Boxers, and Labradors) having an odds ratio of 5.75, 10.67, and 13.77, respectively, as compared to mixed breeds, which are similar to the ones reported by previous authors [2,24]. This result could be attributed to either homogeneity of breed among the European canine populations and/or to the presence, in our country, of the same predisposing factors (environment and food) capable of modifying the microbiota and representing the trigger of the disease.

Similarly, the gender-related prevalence in the development of the disease, comparing the present data with those obtained from the admission of the canine population to the hospital in the same period under examination, indicated a significant higher prevalence of the disease in males vs. females, in contrast to what had already been reported by some authors [6,24], who excluded a polarization of the disease based on sex.

Again, the age of the dog population included in the present study, (37 (7–168) and 46 (9–165) months, for the dogs in Group A and B, respectively, without differences between the groups) was in line with previous studies [2,10], involving middle-aged dogs with IRE and NRE.

On the other hand, these results were in contrast with what had already been reported by Okanishi et al. [25] in a population of Shiba dogs which were older dogs with IBD, with a negative prognosis as compared to those who had controlled IBD. In any case, it should be pointed out that Okanishi’s study and the present study differed in sample size (25 dogs in Okanishi’s work vs. 92 dogs in this study) and in the breeds involved (only Shiba dogs in the first vs. different breeds in this study). Above all, this latter variable could have influenced the different results between the studies.

In summary, these data suggest that the disease, as regards signalment of the canine population involved, and etiology, which, one must remember, is presumed to be multifactorial (genetic predisposition, interactions between food components, environmental factors, and changes in the intestinal microbiota [26]), did not differ in Italy as compared to the rest of Europe [6,24].

Certainly, more interesting is the result of the multifactorial analysis which incorporated anamnestic, clinical, and laboratory data and which, in summary, provided a significant difference only for the total proteins (with a minor value in the dogs in Group B), and previous use of corticosteroids (more frequently already utilized in dogs of Group B), as compared to the total amount of data entered.

Again, the finding of a lower serum protein concentration in the group of dogs which did not respond to the treatment was not surprising since hypoalbuminemia (<20 g/L) is already known to be related to a negative outcome or an absent response to treatment [2,6,27], and protein-losing enteropathy is considered to be life ending in 54.2% of dogs with chronic enteropathy [28].

The cause of the hypoproteinemia, identified in the Group B dogs, could have been due to the chronicity and severity of the inflammatory process, which is capable of determining malabsorption or lack of protein due to lymphangiectasis or crypt disease.

Certainly of greater importance, and novelty, was the correlation between the negative prognosis and the previous use of corticosteroids in the year prior to the time of inclusion which emerged from the multivariate analysis

It could be hypothesized that the dogs belonging to Group B had more severe signs and were referred due to poor response to treatment; however, this does not correspond to the inclusion criteria which excluded patients already diagnosed and treated for IRE with immunosuppressant therapy.

Otherwise, it could be hypothesized that treatment with corticosteroids carried out continuously for a period greater than 10 days in the year preceding diagnosis, could have induced an upregulation of *lamina propria* P-gp, which was capable of limiting the absorption of the immunosuppressive drugs, resulting in their low bioavailability, as has already been reported by Allenspach et al. [10].

In fact, the upregulation of *lamina propria* and epithelial P-gp expression in the small intestine in IRE dogs, with respect to healthy dogs which show no detectable P-gp expression in the duodenal epithelium, has been demonstrated not only by Allenspach et al. [10], but also by Van der Heyden et al. [11].

In an analogous way, these data also indicate an upregulation of P-gp expression in the duodenal biopsies of each group, in both the *lamina propria* (35/66 and 11/14 samples of the dogs in Group A and Group B, respectively) and in the epithelium (29/66 and 8/14 samples of the dogs in Group A and Group B, respectively); however, without statistical differences between the groups.

Similar to what has already been documented in human medicine [29,30], Allenspach et al. [10] pointed out how corticosteroids in dogs were able to upregulate P-gp expression; however, Van der Heyden et al. [11] noted that the upregulation of small bowel mucosal P-gp activity may not depend on induction by corticosteroids since it was also evident in IRE dogs not previously treated with glucocorticoids. Similar to what was reported by Van der Heyden et al. [11], in the present study, it was not possible to identify a significant correlation between P-gp expression in the duodenal *lamina propria* and the epithelium and the previous administration of corticosteroids. This could suggest, in disagreement with Allenspach et al. [10], that the induction of P-gp documented in the dogs in the present paper was not exclusively related to previous glucocorticoid treatment.

Furthermore, an increase in duodenal P-gp expression regarding Crohn’s disease in children, as reported by Fakhoury et al. [31] may represent an adaptive mechanism, compensating for reduced P-gp expression and activity in the colon. Unfortunately, in the present study, it was not possible to explore this hypothesis due to the lack of P-gp analysis of colonic biopsy samples.

Therefore, another element which could justify the dogs belonging to the non-responsive group (Group B) rather than to the responsive group (Group A) could be a diagnostic error, diagnosing what actually was a neoplastic disease (intestinal lymphoma) as inflammatory (IRE) [32,33].

In fact, histological examination of the endoscopic samples could make it difficult to differentiate inflammatory enteropathy from low-grade lymphoma, suggesting how low-grade lymphoma could be underestimated in dogs [34].

The Group B samples underwent additional immunophenotyping and clonality testing determinations to confirm the histological diagnosis.

The samples examined by IHC (16/19) showed the presence of CD3-positive and CD79-positive lymphocytes, demonstrating a coexistence of T- and B-lymphocytes, respectively, thus confirming the histological diagnosis of IRE. The results of the clonality testing carried out on the same biopsy samples analyzed for the IHC instead showed that only 9 of 16 cases were polyclonal for BCR and TCR genes, meanwhile the other samples were clonal (5/16) or oligoclonal (2/16) for TCR and/or BCR genes.

Similar non-consensual results characterized the study of Luckschander-Zeller et al. [33] in which the authors had tried to explain this mismatch of results as being due to the high sensitivity of the clonality test which was capable of detecting latent intestinal lymphoma not detectable with histology, rather than being due to a monoclonal rearrangement occurring in a chronic inflammatory process. The same authors then subsequently moved towards the second hypothesis based on the finding of the absence of intestinal lymphoma in a 10-week follow-up.

In the present case, the dogs in Group B, unlike those of the previous paper [33], did not respond to any treatment. In addition, the dogs were followed up longer than the follow-up reported by Luckschander-Zeller et al. [33]. Accordingly, in the present study, it cannot be affirmed with certainty that the clonal pattern shown by 5/16 dogs depended on a clonal rearrangement in a chronic inflammatory process, rather than on a latent intestinal lymphoma.

## 5. Conclusions

In conclusion, factors related to the development of non-responsive enteropathy are represented by previous treatment with steroids and hypoproteinemia.

Unlike what had previously been reported in the literature, analysis of the present results does not allow attributing the lack of response to therapeutic treatment to an activation of P-gp in lymphocytes infiltrating the intestinal *lamina propria* and the epithelium, since the activation of P-gp characterizes both the group of responsive dogs and that of unresponsive dogs; therefore, it does not represent a useful element for prognosis.

Moreover, the expression of P-gp is not linked to a previous treatment with corticosteroids, so other factors can influence the activation of this protein.

Furthermore, in the group of non-responsive dogs, approximately a quarter of them showed a clonal response for T lymphocytes, of which in one case a concomitant rearrangement for BCR was evidenced, although with histology and immunohistochemistry, they were not suggestive of lymphoma.

Additional studies are needed to understand whether these aspects shown by clonality testing are linked to chronic reactivity in the course of severe inflammation or rather anticipate the development of a lymphoma.

## Figures and Tables

**Figure 1 animals-11-02315-f001:**
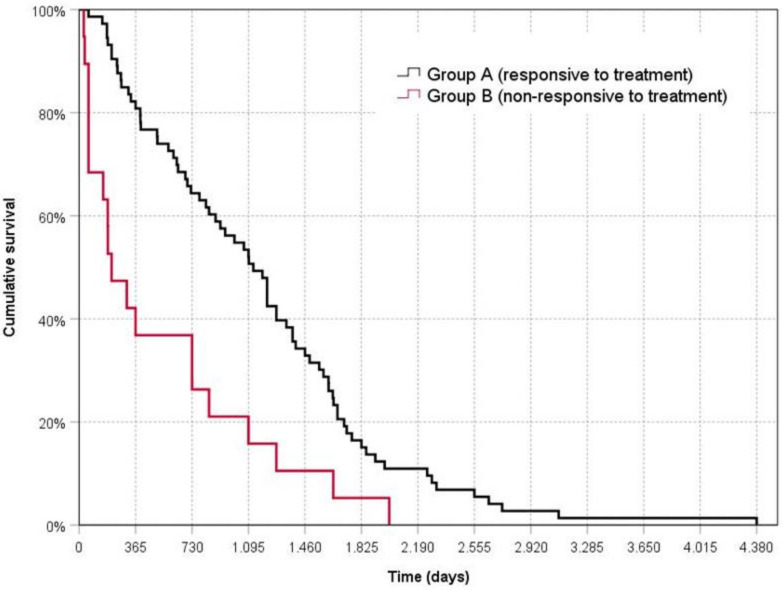
Kaplan––Meier survival curve for dogs responsive or non-responsive to treatment. Time zero represents the date of diagnosis.

**Figure 2 animals-11-02315-f002:**
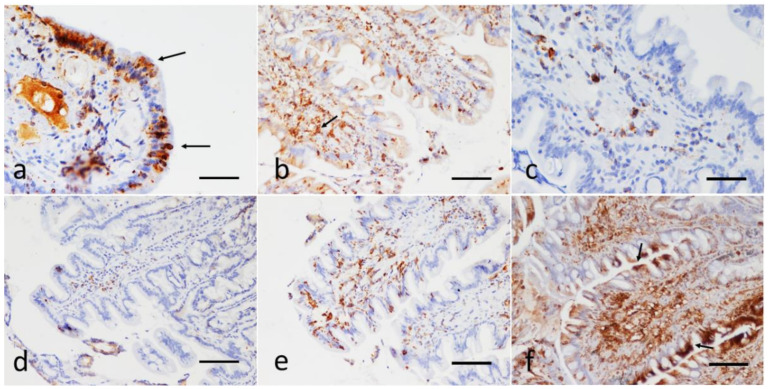
Sections of the duodenum from dogs with IRE. Immunohistochemical evaluation of the epithelial and the *lamina propria* infiltrating lymphocytes; P-gp antibody immunohistochemistry, Streptavidin-biotin immunoperoxidase technique (brown stain). (**a**) Focal, strong cytoplasmic expression of P-gp in enterocytes, reinforced on the apical luminal portion (arrows), scale bar: 100 µm. (**b**) Diffuse, mild, with focal expression at the tips of the villi. Note the strong positivity also in numerous submucosal lymphocytes (arrow), scale bar: 300 µm. (**c**,**d**) Infiltration of the duodenal *lamina propria* with a scarce number of P-gp positive lymphocytes (score 2). There is no expression of P-gp at the enterocytes; scale Bar: 100 µm. (**e**) Strong P-gp positivity in a moderate number of *laminae propria* lymphocytes (score 3); scale bar: 300 µm. (**f**) Strong and diffuse P-gp expression in the lymphocyte submucosal infiltrate (score 4). Note the marked focal positivity also in the apical portion of the duodenal epithelial cells (arrows); scale bar: 300 µm.

**Table 1 animals-11-02315-t001:** Crude odds ratio (OR) of IRE development by sex and breed.

				Odds Ratio		
Category		No. of Cases	Control	Point Estimate	95% Confidence Interval	*p*
Sex ^a^	Male	63	53,950	2.17	1.40–3.37	<0.001
	Female	29	53,958			
Breed ^b^	German Shepherd	13	2917	5.75	2.86–11.58	<0.001
	Boxer	12	1452	10.67	5.21–21.87	<0.001
	Labrador	5	469	13.77	5.14–36.83	<0.001
	Mixed Breed	20	25,825			
Sex × breed	Male German Shepherd	10	1713	2.34	0.64–8.53	0.18
	Female German Shepherd	3	1204			
	Male Boxer	8	797	1.64	0.49–5.48	0.41
	Female Boxer	4	655			
	Male Labrador	5	265	8.47	0.47–154.11	0.13
	Female Labrador	0	204			
	Male Mixed Breed	13	12,380	2.02	0.80–5.06	0.13
	Female Mixed Breed	7	13,445			

^a^ reference category: all the females at the first access to the hospital; ^b^ reference category: mixed breed.

**Table 2 animals-11-02315-t002:** Results of the univariate analysis.

Variables	Categories	Group A (Responsive)	Group B (Non-Responsive)	Total	*p*
Sex	Male	52	11	63	0.26
	Female	21	8	29	
	total	73	19	92	
Diet previously utilized	homemade	12	3	15	0.89
	commercial	38	11	49	
	mixed diet	23	5	28	
	total	73	19	92	
Previous enteric parasites, protozoal and/or parvovirus infection	yes	26	4	30	0.23
	no	47	15	62	
	total	73	19	92	
Previous treatment with glucocorticoids	yes	13	10	23	<0.01 ^a^
	no	60	9	69	
	total	73	19	92	
Decreased appetite	yes	18	10	28	0.02 ^a^
	no	55	9	64	
	total	73	19	92	
Weight loss	yes	36	14	50	0.06 ^a^
	no	37	5	42	
	total	73	19	92	
Vomiting	yes	40	5	54	0.03 ^a^
	no	33	14	38	
	total	73	19	92	
Diarrhea	yes	66	19	85	0.34
	no	7	0	7	
	total	73	19	92	
Ascites or peripheral edema	yes	5	4	9	0.08 ^a^
	no	68	15	83	
	total	73	19	92	
CCECAI	0–4	38	10	48	0.85
	>5	31	9	40	
	total	69	19	88	
PCV (%)	37–55	59	12	71	<0.01 ^a^
	<37	2	5	7	
	>55	6	1	7	
	total	67	18	85	
Platelet count (×10^3^/μL)	160–550	58	14	72	0.40
	<160	2	1	3	
	>550	7	4	11	
	total	67	19	86	
White blood cells (×10^3^/μL)	6–17	59	15	74	0.42
	<6	1	0	1	
	>17	7	4	11	
	total	67	19	86	
Serum cholesterol (mg/dL)	≥140	47	7	54	<0.01 ^a^
	<140	12	10	22	
	Total	59	17	76	
Serum albumin (g/dL)	≥2	52	8	60	<0.01 ^a^
	<2	15	11	26	
	total	67	19	86	
Serum total protein (g/dL)	≥5.6	50	4	54	<0.001 ^a^
	<5.6	18	15	33	
	total	68	19	87	
Serum cobalamin (ng/L)	250–730	25	4	29	0.19
	<250	16	3	19	
	>730	4	3	7	
	total	45	10	55	
Serum folate (μg/L)	6.5–11.5	13	2	15	0.61
	<6.5	15	5	20	
	>11.5	17	3	20	
	total	45	10	55	
P-gp score in the *lamina propria* lymphocytes	<3	31	3	34	0.08 ^a^
	≥3	35	11	46	
	total	66	14	80	
P-gp score in the epithelial cells	<2	37	6	43	0.37
	≥2	29	8	37	
	total	66	14	80	

^a^ Significant association *p* < 0.10.

**Table 3 animals-11-02315-t003:** Results of logistic regression analysis.

Factors	Dogs Not Responding to Treatment/Total (%)	OR	95% CI for OR	*p*
Previous treatment with glucocorticoids				
yes	10/23 (43.5)			
no	9/69 (8.7)	8.06	1.18–54.82	0.033
PCV (%)				
37–55	12/71 (16.9)			
<37	5/7 (71.4)	0.86	0.04–18.54	0.925
>55	1/7 (14.3)	31.20	0.93–1047.20	0.055
Serum total protein (g/dL)				
≥5.6	4/54 (7.4)			
<5.6	15/33 (45.5)	37.33	3.03–459.80	0.005
P-gp score in the *lamina propria* lymphocyte				
<3	4/10 (40.0)			
≥3	14/20 (70.0)	1.97	0.32–12.15	0.077
Constant		0.002		0.001

**Table 4 animals-11-02315-t004:** Results of the molecular clonality analysis and immunohistochemical examinations for phenotype evaluation, carried out on the biopsy samples of the dogs (16/19) in Group B.

Case	BCR Gene Rearrangement	TCR Gene Rearrangement	CD3 IHC	CD79 IHC
1	/	/	/	/
2	Polyclonal	Polyclonal	+	+
3	Polyclonal	Oligoclonal	+	+
4	Polyclonal	Polyclonal	+	+
5	Polyclonal	Polyclonal	+	+
6	Polyclonal	Polyclonal	+	+
7	Polyclonal	Clonal	+	+
8	Polyclonal	Oligoclonal	+	+
9	Polyclonal	Clonal on a polyclonal background	+	+
10	Polyclonal	Polyclonal	+	+
11	Polyclonal	Polyclonal	+	+
12	/	/	/	/
13	Polyclonal	Polyclonal	+	+
14	/	/	/	/
15	Polyclonal	Clonal on a polyclonal background	+	+
16	Polyclonal	Polyclonal	+	+
17	Polyclonal	Clonal	+	+
18	Polyclonal	Polyclonal	+	+
19	Clonal	Clonal on a polyclonal background	+	+

/: sample not analyzed. +: positive reaction.

## Data Availability

The data presented in this study are available on request from the corresponding author.

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
