# Peer review of "Evaluation of Prognostic Factors, including Duodenal P-Glycoprotein Expression, in Canine Chronic Enteropathy"

_animals, 2021, doi:10.3390/ani11082315_

Round 1

Reviewer 1 Report

Dear authors,

I really enjoyed reading your work, in this sense I leave some contributions that aim to make the reading of the article more expeditious.

Introduction

A latent conflict between the use of the terms IBD and canine chronic enteropathies can be noted in the work, namely in the use of clinical activity indices related to IBD.

Work aims and conclusions

In conclusion, the time of survival in CE dogs is related to previous treatment with steroids and hypoprotidemia. A conclusion not related to a specific objective of the work.

Furthermore, in the group of non-responsive dogs, approximately a quarter of them showed a clonal response for T lymphocytes, of which in 1 case a concomitant rearrangement for BCR was evidenced, although with histology and immunohistochemistry they were not suggestive of lymphoma. 
Additional studies are needed to understand whether these aspects shown by clonality testing are linked to chronic reactivity in the course of severe inflammation or rather anticipate the development of a lymphoma. A conclusion not related to a specific objective of the work.

Materials and Methods

The criteria for inclusion in the study are not clear, as is the control group.

I could not find clearly this statement (refered at Page 12) at the MM section

Page 12, lines 402,403 - "criteria which excluded patients already diagnosed and treated for IRE."

Page 3, Lines 116 to 126 - The scientific language must be objective and clear, in this sense the definition of groups A and B should be rewritten. It is at this point that group A should be defined as the animals that respond and group B as those that do not respond to therapies.

Discussion

Page 12, Line 394 - 396 - I couldn't agree with this paragraph. There are other reasons to losing protein such as crypt disease.

Conclusion

Page 13, Line 463 - How do you prefer hypoprotidemia to hypoproteinemia?

The construction of the conclusions must respect the objectives of the work, which is not verified in the specific case of this work.

"Based on these previous results, the aim of this retrospective single-center trial was to

1) evaluate which factors could predict the response to treatment in dogs with IRE; without answer in conclusions

3) evaluate whether a previous treatment with steroids could induce a higher expression of duodenal P-gp. without answer in conclusions

Author Response

Comments are reported in the attached file

Reviewer 2 Report

I carefully reviewed the manuscript entitled “Evaluation of the prognostic ability of duodenal P-glycoprotein expression in differentiating immunosuppressant responsive vs. non-responsive enteropathy in dogs” by Pietra and colleagues. This is an interesting and well-conceived study that extensively explored the factors at the base of the non-responsive enteropathy in the canine species, with the aim to identify which of them could predict the non-responsiveness of the disease to the treatment.

The topic is interesting, and even if it has already been studied, the present article add some interesting data and points of view that make it suitable for the publication in the Journal Animals.  In my opinion, it needs just some minor revisions.

The most important point is that it is not always clear the main aim of the study. The title clearly stated that the “protagonist” of the study is the P-glycoprotein. However, in various parts of the text, this messages is strongly diluted among all the other factors included in the analysis. For example, the authors spent the first paragraphs of the discussion approaching all the other factors (breed, sex, clinical and biochemical data…) and only in a second moment they discussed the results about the P-glycoprotein expression. In my opinion, the authors should clearly decide which is the main aim and reorganize the manuscript coherently with this decision. I think that the amount of results, their quality, and the way the authors used to discuss each of them is good, thus maybe both the choices can produce an interesting and useful article. However, it is very important to reorganize the article according to a clear aim, and only after to present the other results, which are always interesting.

Other minor flaws are reported here below:

  • Title: the word “ability” doesn’t seem very adequate in this context.
  • Abstract: in my opinion, it is non well balanced, your contributions are limited to just some lines.
  • MM: please provide some more information about the anti-P-glycoprotein antibody? Is was already validate for canine tissues? How? Or did you have to make your own validation? How?
  • Results: I spent some time to understand the result about vomiting proportions in A and B group. In the text (line 264) you stated that the group A presented a lower proportion of vomiting individuals. However, in Table 2, the Group A presented more of 50% “vomiting” dogs (40/73), while the Group B more or less 25% (5/19).
  • Discussion: please see above.
